

# OMI measured increasing SO₂ emissions due to energy industry expansion and relocation in Northwestern China

**Authors:**
Zaili Ling[1], Tao Huang[1,*], Yuan Zhao[1], Jixiang Li[1], Xiaodong Zhang[1], Jinxiang Wang[1], Lulu Lian[1], Xiaoxuan Mao[1], Hong Gao[1], Jianmin Ma[1,2,3,*]

**Affiliations:**
[1]Key Laboratory for Environmental Pollution Prediction and Control, Gansu Province, College of Earth and Environmental Sciences, Lanzhou University, Lanzhou 730000, P. R. China
[2]Laboratory for Earth Surface Processes, College of Urban and Environmental Sciences, Peking University, Beijing, 100871, China
[3]CAS Center for Excellence in Tibetan Plateau Earth Sciences, Chinese Academy of Sciences, Beijing, 100101, China

**Corresponding author:** Jianmin Ma, Tao Huang
College of Earth and Environmental Sciences, Lanzhou University, 222, Tianshui South Road, Lanzhou 730000, China

Email: jianminma@lzu.edu.cn; huangt@lzu.edu.cn





**Abstract**
The rapid economy growth makes China the largest energy consumer and sulphur
dioxide ($SO_2$) emitter in the world. In this study, we estimated the trends and step
changes in the planetary boundary layer (PBL) vertical column density (VCD) of $SO_2$
from 2005 to 2015 over China measured by the Ozone Monitoring Instrument (OMI).
We show that these trends and step change years coincide with the effective date and
period of the national strategy for energy development and relocation in northwestern
China and the regulations in the reduction of $SO_2$ emissions. Under the national
regulations in the reduction $SO_2$ emissions in eastern and southern China, $SO_2$ VCD
in the Pearl River Delta (PRD) of southern China exhibited the largest decline during
2005-2015 at a rate of -7% $yr^{-1}$, followed by the North China Plain (NCP) (-6.7% $yr^{-1}$),
Sichuan Basin (-6.3% $yr^{-1}$), and Yangtze River Delta (YRD) (-6% $yr^{-1}$), respectively.
The Mann–Kendall (MK) test reveals the step change points of declining $SO_2$ VCD in
2009 for the PRD and 2012-2013 for eastern China responding to the implementation
of $SO_2$ control regulation in these regions. In contrast, the MK test and regression
analysis also revealed increasing trends of $SO_2$ VCD in northwestern China,
particularly for several "hot spots" featured by growing $SO_2$ VCD in those large-scale
energy industry parks in northwestern China. The enhanced $SO_2$ VCD is potentially
attributable to increasing $SO_2$ emissions due to the development of large-scale energy
industry bases in energy-abundant northwestern China under the national strategy for
the energy safety of China in the 21st century. We show that these large-scale energy
industry bases could overwhelm the trends and changes in provincial total $SO_2$



emissions in northwestern China and contributed increasingly to the national total SO$_2$
emission in China. Given that northwestern China is more ecologically fragile and
uniquely susceptible to atmospheric pollution as compared with the rest of China,
increasing SO$_2$ emissions in this part of China should not be overlooked and merit
scientific research.
**1. Introduction**

Sulfur dioxide (SO$_2$) is one of the criteria air pollutants emitted from both

anthropogenic and natural sources. The combustions of sulfur-containing fuels, such
as coal and oil, are the primary anthropogenic emitters, which contributed to the half
of total SO$_2$ emissions (Smith et al., 2011; Lu et al., 2010; Stevenson et al., 2003;
Whelpdale et al., 1996).With the rapid economic growth in the past decades, China
has become the world's largest energy consumer accounting for 23% of global energy
consumption in 2015 (BIEE, 2016). Coal has been a dominating energy source in
China and accounted for 70% of total energy consumption in 2010 (Kanada et al.,
2013). The huge demand to coal and its high sulfur content make China the largest
SO$_2$ emission source in the world (Krotkov et al., 2016; Su et al., 2011), which also
accounted for two-third of Asia's total SO$_2$ emission (Ohara et al., 2007). From 2000
to 2006, the total SO$_2$ emission in China increased by 53% at an annual growth rate of
7.3% (Lu et al., 2010). To reduce SO$_2$ emission, from 2005 onward the Chinese
government has issued and implemented a series of regulations, strategies, and SO$_2$
control measures, leading to a drastic decrease of SO$_2$ emission, particularly in eastern





and southern China (Lu et al., 2011; Li et al., 2010).

Recently, two research groups led by NASA (National Aeronautics and Space

Administration) and Lanzhou University of China published almost simultaneously
the temporal and spatial trends of $SO_2$ in China from 2005 to 2015 using the OMI
retrieved $SO_2$ PBL column density after the OMI is lunched for 11 years (Krotkov et
al., 2016; Shen et al., 2016). The results reported by the two groups revealed
widespread decline of $SO_2$ in eastern China for the past decade. Shen et al noticed,
however, that, in contrast to dramatic decreasing $SO_2$ emissions in densely populated
and industrialized eastern and southern China, the OMI measured $SO_2$ in northwestern
China appeared not showing a decreasing trend. This is likely resulted from the
energy industry relocation and development in energy-abundant northwestern China
in the past decades under the national strategy for China's energy development and
safety during the 21st century. Concern is raised for the potential impact of $SO_2$
emissions on the ecological environment and health risk in northwestern China
because high $SO_2$ emissions could otherwise damage the rigorous ecological
environment in this part of China, featured by very low precipitation and sparse
vegetation coverage which reduce considerably the atmospheric removal of air
pollutants (Ma and Xu, 2017).

To assess and evaluate the risks of the ecological environment and public to the

growing $SO_2$ emissions in northwestern China, it is necessary to investigate the
spatiotemporal distributions of $SO_2$ concentrations and emissions. However, the
ground measurements of ambient $SO_2$ are scarce temporally and spatially in China,



and often subject to large errors and uncertainties. Owing to the rapid progresses in
the remote sensing techniques, satellite retrieval of air pollutants has become a
powerful tool in the assessment of emissions and spatiotemporal distributions of air
pollutants. In recent several years, OMI (Dutch Space, Leiden, The Netherlands,
embedded on Aura satellite) retrieved $SO_2$ column concentrations have been
increasingly applied to elucidate the spatiotemporal variation of global and regional
$SO_2$ levels and its emissions from large point sources, and evaluate the effectiveness
of $SO_2$ control policies and measures (Krotkov et al., 2016; McLinden et al., 2015,
2016;Ialongo et al., 2015; Fioletov et al., 2015, 2016;Wang et al., 2015;Li et al.,
2010). The decadal operation of the OMI provides the relatively long-term $SO_2$ time
series data with high spatial resolution which are particular useful for assessing the
changes and trends in $SO_2$ emissions induced by national regulations and strategies.
The present study aims to (1) assess the spatiotemporal variations of $SO_2$ and its trend
under the national strategy for energy industry development in northwestern China by
making use of the OMI-measured $SO_2$ data during 2005-2015; (2) to further examine
the usefulness of the satellite remote sensing of air quality.

**2 Data and methods**
**2.1 Satellite data**
We collected the level 3 OMI daily planetary boundary layer (PBL) $SO_2$ vertical
column density (VCD) data in Dobson units (1 DU=$2.69\times10^{16}$ molecules cm$^{-2}$)
produced by the principal component analysis (PCA) algorithm (Li et al., 2013). The



spatial resolution is 0.25°×0.25° latitude/ longitude, available at Goddard Earth Sciences
Data               and               Information               Services               Center
(http://disc.sci.gsfc.nasa.gov/Aura/data-holdings/OMI/omso2_v003.shtml). This algorithm
yields one-step $SO_2$ VCD. However, as Fioletov et al (2016) noted, the PCA retrieved $SO_2$
VCD was virtually derived by adoption of an effective air mass factor (AMF) of 0.36 which
is best applicable in the summertime in the eastern United States (US). The algorithm may
cause systematic errors if anthropogenic emission sources are located in different latitudes
and under complex topographic and underlying surface conditions. For instance, Wang
(2014) has shown that AMF≈0.57 in eastern China. In the present study, we have adopted
the AMFs values in China provided by Fioletov et al (2016) to adjust OMI measured VCD
in the estimation of the $SO_2$ emission burden of major point sources in northwestern China.
**2.2 $SO_2$ monitoring, emission, and socioeconomic data**
**Figure 1** is a China map which highlights 6 provinces in northwestern China,
including Shaanxi, Gansu, Qinghai, Ningxia, Xinjiang, and Inner Mongolia.
Traditionally, Inner Mongolia is not classified as a northwestern province in China.
Given that the most energy resources in Inner Mongolia are located in its western
part of this province (Fig. 1), here we include this province in northwestern China.
North China Plain (NCP), Beijing-Tianjin-Hebei (BTH), Yangtze River Delta (YRD),
Pearl River Delta (PRD), and Sichuan Basin are also shown in the map. To evaluate
and verify the spatial $SO_2$ VCD from OMI, we collected ground $SO_2$ monitoring data
of 2014 through 2015 at 188 sampling sites (cities) across China (**Figure 1**),
operated by the National Environmental Monitoring Center, available at



http://www.aqistudy.cn/historydata. The statistics between OMI retrieved $SO_2$ VCD
and monitored monthly and annually averaged $SO_2$ air concentrations during
2014-2015 at 188 operational air quality monitoring stations across China are
presented in **Table S1** of Supplement. **Figure S1** is the correlation diagram between
$SO_2$ VCD and sampled data. As shown in **Table S1** and **Fig. S1**, the OMI measured
$SO_2$ VCDs agree well with the monitored ambient $SO_2$ concentrations across China
at the correlation coefficient of 0.85 ($p < 0.05$) (**Table S1**). **Figure 2** further compared
annually averaged $SO_2$ VCDs and $SO_2$ air concentrations from 2005 to 2015 in 6
capital cities in Urumqi (Xinjiang), Yinchuan (Ningxia), Beijing (BTH and NCP),
Shanghai (YRD), Guangzhou (PRD), and Chongqing (Sichuan Basin), respectively.
The mean $SO_2$ concentration data were collected from provincial environmental
bulletin published by the Ministry of Environmental Protection of China (MEPC)
(http://www.zhb.gov.cn/hjzl/zghjzkgb/gshjzkgb. Results show that the annual
variation of mean $SO_2$ VCDs match well with the monitored data except for Urumqi,
the capital of Xinjiang Uygur Autonomous region. The OMI retrieved $SO_2$ VCDs in
Shanghai and Chongqing are higher than the measured $SO_2$ concentrations from
2010 to 2015 but the both show consistent temporal fluctuation and trend. The
measured $SO_2$ concentrations peaked in 2013 in Yinchuan whereas the $SO_2$ VCD
reached the peak in 2012 and decreased thereafter. OMI measured $SO_2$ VCDs in
Urumqi show different yearly fluctuations compared with its annual concentrations.
The measured $SO_2$ concentrations in Urumqi decreased from 2011 to 2015 whereas
the OMI measured $SO_2$ VCDs did not illustrate obvious changes. It is not clear the





causes leading to such the inconsistence. Measured concentrations might be subject
to errors or not properly reported. Since the monitored $SO_2$ concentrations were
collected in the urban area spatially averaged over 8 monitoring sites across the city
whereas the OMI measured $SO_2$ VCD was averaged over all model grid points
($0.25\times0.25$ latitude/longitude resolution) in Urumqi city. This could also result in the
inconsistence between $SO_2$ VCD and measured data. However, such the error
appeared not occurring in other cities.
$SO_2$ anthropogenic emission inventory in China with a $0.25°$ longitude by $0.25°$
latitude resolution for every two years from 2008 to 2012 was adopted from Multi
resolution Emission Inventory for China (MEIC) (Li et al., 2017, available at
http://www.meicmodel.org). The comparison between annual OMI $SO_2$ VCD and $SO_2$
emissions in China is presented in **Fig. 3**. As shown, the annual variation in $SO_2$
VCDs also agrees reasonably well with $SO_2$ emission data except for Midong. The
OMI measured $SO_2$ VCD in the PRD and Sichuan Basin decreased from 2008 to 2012
but $SO_2$ emission changed little. Compared with the other five marked regions, the
satellite measured $SO_2$ VCD in Midong declined in 2010 and inclined in 2012.
However, $SO_2$ emissions in Midong increased in 2012 at about factor of 11 and 8
higher than that in 2008 and 2010. It should be noted that the MEIC $SO_2$ emission
inventory from the bottom-up approach might be subject to large uncertainties due to
the lack of sufficient knowledge in human activities and emissions from different
sources (Li et al., 2017; Zhao et al., 2011; Kurokawa et al., 2013). From this
perspective, the satellite remote sensing provides a powerful tool in monitoring $SO_2$





emissions from large point sources and the verification of emission inventories
(Fioletov et al., 2016; Wang et al., 2015).
The socioeconomic data were collected from the China Statistical Yearbooks and
China Energy Statistical Yearbook, published by National Bureau of Statistics of
China (NBSC) (http://www.stats.gov.cn/tjsj/ndsj/;http://tongji.cnki.net/kns55/Navi/
HomePage.aspx?id=N2010080088&name=YCXME&floor=1), as well as China
National Environmental Protection Plan in the Eleventh Five-Years (2006-2010) and
Twelfth Five-Years (2011-2015) released by MEPC (http://www.zhb.gov.cn).
**2.3 Trends and step change**
The long-term trends of $SO_2$ VCD were estimated by linear regressions of the
gridded annually $SO_2$ VCD against their time sequence of 2005 through 2015. The
gridded slopes (trends) of the linear regressions denote the increasing (positive) or
decreasing (negative) rates of $SO_2$ VCD (Wang et al., 2016; Huang et al., 2015;
Zhang et al., 2015, 2016).
The Mann-Kendall (MK) test was also employed in the assessments of the
temporal trend and step change point year of $SO_2$ VCD time series. The MK test is a
nonparametric statistical test (Mann,1945; Kendall, 1975), which is useful for
assessing the significance of trends in time series data (Waked et al., 2016; Fathian et
al., 2016). The MK test is often used to detect a step change point in the long term
trend of a time series dataset (Moraes et al, 1998; Li et al., 2016; Zhao et al., 2016).
It is suitable for non-normally distributed data and censored data which are not
influenced by abnormal values (Yue and Pilon, 2004; Sharma et al 2016; Yue and





Wang., 2004; Gao et al 2016; Zhao et al., 2015). Recently, MK-test has also been
used in trend analysis for the time series of atmospheric chemicals, such as persistent
organic pollutants, surface ozone ($O_3$), and non-methane hydrocarbon (Zhao et al.,
2015; Assareh et al.,2016; Waked et al.,2016; Sicard et al., 2016). Here the MK test
was used to identify the temporal variability and step change point of $SO_2$ VCD for
2005-2015 which may be associated with the implementation of the national strategy
and regulation in energy industry development and emission control during this
period of time. Under the null hypothesis (no trend), the test statistic was determined
using the following formula:
$$S_k = \sum_{i=1}^{k} r_i \ (k= 2, 3, ...,n) \tag{1}$$

where $S_k$ is a statistic of the MK test, and
$$r_i = \begin{cases} +1,(x_i > x_j) \\ 0,(x_i \le x_j) \end{cases} \quad (j=1,2, ...,i\text{-}1) \tag{2}$$

where $x_i$ is the variable in time series $x_1$, $x_2$, ..., $x_i$, $r_i$ is the cumulative number for
$x_i > x_j$. The test statistic is normally distributed with a mean and variance given by:
$$E(S_k) = k(k-1)/4 \tag{3}$$

$$Var(S_k) = \frac{k(k-1)(2k+5)}{72} \tag{4}$$

From these two equations one can derive a normalized $S_i$, defined by
$$UF_k = \frac{S_k - E(S_k)}{\sqrt{Var(S_k)}} \quad (k=1, 2, ...,n) \tag{5}$$

where $UF_k$ is the forward sequence, the backward sequence $UB_k$ is calculated using
the same function but with the reverse data series such that $UB_k$=-$UF_k$.



In a two-sided trend test, a null hypothesis is accepted at the significance level if
$\left|(UF_k)\right| \leq (UF_k)_{1-\alpha/2}$, where $(UF_k)_{1-\alpha/2}$ is the critical value of the standard normal
distribution, with a probability of $a$. When the null hypothesis is rejected (i.e., when
any of the points in $UF_k$ exceeds the confidence interval ±1.96; P=0.05), an significant
increasing or decreasing trend is determined. $UF_k > 0$ often indicates an increasing
trend, and vice versa. The test statistic used in the present study enables us to
discriminate the approximate time of trend and step change by locating the
intersection of the $UF_k$ and $UB_k$ curves. The intersection occurring within the
confidence interval (-1.96, 1.96) indicates the beginning of a step change point
(Moraeset al., 1998; Zhang et al., 2011; Zhao et al., 2015).

**3 Results and discussion**
**3.1. Spatiotemporal variation in OMI measured SO₂**
Given higher population density and stronger industrial activities, eastern and
southern China are traditionally industrialized and heavily contaminated regions by
air pollutions and acid rains caused by SO₂ emissions. **Figure 4a** shows annually
averaged OMI SO₂ VCD over China on a 0.25 ° × 0.25 ° latitude/longitude
resolution averaged from 2005 to 2015. SO₂ VCD was higher considerably in eastern
and central China, and Sichuan Basin than that in northwestern China. The highest
SO₂ VCD was found in the NCP, including Beijing-Tianjin-Hebei (BTH), Shandong,
and Henan province. The annually averaged SO₂ VCD between 2005-2015 in this
region reached 1.36 DU. This result is in line with previous satellite remote sensing





retrieved $SO_2$ emissions in eastern China (Krotkov et al 2016; Lu et al., 2010;
Bauduin et al., 2016; Jiang et al 2012; Yan et al., 2014). However, in contrast to the
spatial distribution of decadal mean $SO_2$ VCD (**Fig. 4a**), the slopes of the linear
regression relationship between annual average OMI-retrieved $SO_2$ VCD and the
time sequence from 2005 to 2015 over China show that the negative trends
overwhelmed industrialized eastern and southern China, particularly in the NCP,
Sichuan Basin, the YRD, and PRD, manifesting significant decline of $SO_2$ emissions
in these regions. $SO_2$ VCD in the PRD exhibited the largest decline at a rate of 7%
$yr^{-1}$, followed by the NCP (6.7% $yr^{-1}$), Sichuan Basin (6.3% $yr^{-1}$), and the YRD (6%
$yr^{-1}$), respectively. Annual average $SO_2$ VCD in the PRD, NCP, Sichuan Basin, and
YRD decreased by 52%, 50% , 48%, and 46% in 2015 compared to 2005 (**Fig. 5**),
though the annual fluctuation of $SO_2$ VCD shows rebounds in 2007 and 2011 which
are potentially associated with the economic resurgence stimulated by the central
government of China (He et al., 2009; Diao et al., 2012). The reduction of $SO_2$ VCD
after 2011 in these regions reflects virtually the response of $SO_2$ emissions to the
regulations in the reduction of $SO_2$ release, the mandatory application of the flue-gas
desulfurization (FGD) on coal-fired power plants and heavy industries, and the
slowdown in the growth rate of the Chinese economy (CSC, 2011a; Wang et al.,
2015, Chen et al., 2016).
As also shown in **Fig. 4b**, in contrast to widespread decline of $SO_2$ VCD, there
are two "hot spots" featured by moderate increasing trends of $SO_2$ VCD, located in
the China's Energy Golden Triangle (EGT, Shen et al., 2016, Ma and Xu, 2017) and



Urumqi-Midong regions in northwestern China. The annual growth rate of $SO_2$ VCD
from 2005 to 2015 are 3.4% $yr^{-1}$ in the EGT and 1.8% $yr^{-1}$ in Urumqi-Midong,
respectively (**Fig. 4b**). Further details are presented in **Table 1**. $SO_2$ VCDs in these
two regions peaked in 2011 and 2013 which were 1.6 and 1.7 times of that in 2005
(**Fig. 5**). The raising $SO_2$ VCDs in the part of the EGT have been reported by Shen et
al. (2016). The second hot spot is located in Midong industrial park, about 40 km
away from Urumqi, the capital of the Xinjiang Uygur Autonomous Region. The both
EGT and Midong industrial parks are featured by extensive coal mining, thermal
power generation, coal chemical, and coal liquefaction industries. The reserve of
coal, oil and natural gas in the EGT is approximately $1.05 \times 10^{12}$ ton of standard coal
equivalent, accounting for 24% of the national total energy reserve in China
(CRGECR, 2015). It has been estimated that there are deposits of 20.86 billion tons
of oil, 1.03 billion cubic meters of natural gas, and 2.19 trillion tons of coal in
Xinjiang, accounting for 30%, 34% and 40% of the national total (Dou, 2009). Over
the past decades, a large number of energy-related industries have been constructed
in northwestern China, such as the EGT and Midong chemical industrial parks in
order to enhance China's energy security in the 21st century and speed up local
economy. Rapid development of energy and coal chemical industries in Ningxia Hui
Autonomous region and Xinjiang of northwestern China alone resulted in the
significant demands to coal mining and coal products. The coal consumption,
thermal power generation, and the gross industrial output increased by 2.7, 3.5, and
6.6 times in Ningxia from 2005 to 2015, and by 2.7, 4.2 and 6.6 times in Xinjiang





during the same period (NBSC, 2005, 2015). As a result, $SO_2$ emissions increased
markedly in these regions, as shown by the increasing trends of $SO_2$ VCD in the
EGT and Midong (**Fig. 4b**). **Figure 6** illustrates the fractions of OMI measured
annual $SO_2$ VCD and $SO_2$ emissions averaged over the 6 provinces of northwestern
China in the annual national total VCD (**Fig. 6a**) and emissions (**Fig. 6b**) from 2005
to 2015. The both $SO_2$ VCD and emission fractions in northwestern China in the
national total increased over the past decade. By 2015, the mean $SO_2$ VCD fraction
in 6 northwestern provinces has reached 38% in the national total. The mean
emission fraction was about 20% in the national total. It should be noted that there
were large uncertainties in provincial $SO_2$ emission data which often underestimated
$SO_2$ emissions from major point sources (Li et al., 2017; Han et al., 2007). In this
sense, OMI retrieved $SO_2$ VCD fraction provides a more reliable estimate to the
contribution of $SO_2$ emission in northwestern China to the national total.

The annual percentage changes in $SO_2$ VCD from 2005 onward are consistent

well with per capita $SO_2$ emissions in China (**Fig. 7**). As aforementioned, while the
annual total $SO_2$ emissions in the well developed BTH, YRD, and PRD were higher
than that in northwestern provinces, the per capita emissions in all provinces of
northwestern China, especially in Ningxia and Xinjiang, were about factors of 1 to 6
higher than that in the BTH, YRD, and PRD, as shown in **Fig. 7**. In contrast to
declining annual emissions from the BTH, YRD, and PRD, the per capita $SO_2$
emissions in almost all western provinces have been growing from 2005 onward.
**3.2 Trend and step changes in OMI measured $SO_2$ by MK test**



285  Given that in the MK test the signs and fluctuations of $UF_k$ are often used to

286  predict the trend of a time series, this approach is further applied to quantify the trends

287  and step changes in annually $SO_2$ VCD time series in those highlighted regions (a-f)

288  in **Fig. 4b** from 2005 to 2015. Results are illustrated in **Fig. 8**. As shown, the forward

289  and backward sequences $UF_k$ and $UB_k$ intersect at least once from 2005 to 2015.

290  These intersections are all well within the confidence levels between -1.96 and 1.96 at

291  the statistical significance $\alpha=0.01$. A common feature of the forward sequence $UF_k$ in

292  eastern and southern China provinces is that $UF_k$ has been declining and become

293  negative from 2007 to 2009 onward (**Fig. 8a-d**), confirming the downturn of $SO_2$

294  atmospheric emissions and levels in these industrialized and well developed regions in

295  China. In the EGT and Midong areas of northwestern China (**Fig. 4b**), however, the

296  $UF_k$ values for $SO_2$ VCD are positive and growing, illustrating clear upward trends of

297  $SO_2$ VCD over these two large-scale energy industry parks, revealing the response of

298  $SO_2$ emissions to the energy industry relocation and development in northwestern

299  China. To guarantee the national energy security and to promote the regional

300  economy, the EGT energy program has been accelerating since 2003 under the

301  national energy development and relocation plan (Zhu and Ruth, 2015; Chen et al.,

302  2016), characterized by the rapid expansion of the Ningdong energy and chemical

303  industrial base (NECIB) which is located about 40 km away from Yinchuan, the

304  capital of Ningxia (Shen et al., 2016). By the end of 2010, a large number of coal

305  chemical industries, including the world largest coal liquefaction and thermal power

306  plants, have been built and operated, and the total installed capacity of thermal power





generating units has reached 1.47 million kilowatts (Zhao, 2016). Under the same
national plan, the Midong industrial park in Xinjiang started to construction and
operation from the early to mid-2000s which has almost the same industrial structures
as those in the EGT, featured by coal-fired power generation, coal chemical industry,
and coal liquefaction.

For those regions with declining trends of $SO_2$ VCD, their step change points in

the NCP, YRD and Sichuan Basin occurred between 2012 and 2013. These step
change points coincide with the implementation of the new Ambient Air Quality
Standard in 2012, which set a lower ambient $SO_2$ concentration limit in the air (MEPC,
2012), and the Air Pollution Prevention and Control Action Plan in 2013 by the State
Council of China (CSC, 2013a). This Action Plan requests to take immediate actions
to control and reduce air pollution in China, including cutting down industrial and
mobile emission sources, adjusting industrial and energy structures, and promoting
the application of clean energy in the BTH, YRD, PRD and Sichuan Basin. The step
change in $SO_2$ VCD over the PRD occurred in the earlier year of 2009-2010 and from
this period onward the decline of $SO_2$ VCD speeded up, as shown by the forward
sequence $UF_k$ which became negative since 2007 and was below the confidence level
of -1.96 after 2009, suggesting significant decreasing VCD from 2009 (**Fig. 8c**). In
April 2002, the Hong Kong Special Administrative Region (HKSAR) Government
and the Guangdong Provincial Government reached a consensus to reduce, on a best
endeavor basis, the anthropogenic emissions of $SO_2$ by 40% in the PRD by 2010,
using        1997        as        the        base        year





(http://www.epd.gov.hk/epd/english/action_blue_sky/files/exsummary_e.pdf). By the
end of 2010, all thermal power units producing more than 0.125 million kilowatts
electricity in the PRD were equipped with the FGD. During the 11th Five-Year Plan
(2006-2010), the thermal power units with 1.2 million kilowatts capacity have been
shut down. $SO_2$ emission was reduced by 18% in 2010 compared to that in 2005
(NBSC, 2006, 2011). This likely caused the occurrence of the step change in $SO_2$
VCD over 2009-2010.
The statistical significant step change points of $SO_2$ VCD in the EGT and
Midong took place in 2006 and 2009, differing from those regions with decreasing
trends of $SO_2$ VCD in eastern and southern China. The first step change point in
2006-2007 corresponds to the increasing $SO_2$ emissions in these two large-scale
energy bases till their respective peak emissions in EGT (2007) and Midong (2008).
The second step change point in 2009 coincides with the global financial crisis in
2008 which slowed down considerably the economic growth in 2009 in China,
leading to raw material surplus and the remarkable reduction in the demand to coal
products.
**3.3 OMI $SO_2$ time series and step change point year in northwestern China**
Since almost all large-scale coal chemical, thermal power generation, and coal
liquefaction industries were built in energy-abundant and sparsely populated
northwestern China over the past two decades, particularly since the early 2000s,
those large-scale industrial parks and bases in this part of China likely play an
important role in the growing $SO_2$ emissions in northwestern provinces. We further



examine the OMI retrieved $SO_2$ VCD to confirm and evaluate the changes in $SO_2$
emissions in northwestern China which should otherwise respond to these
large-scale energy programs under the national plan for energy relocation and
expansion. **Figure 9** displays the MK test statistics for $SO_2$ VCD in the 6 provinces
in northwestern China from 2005-2015. The forward sequence $UF_k$ suggests
decreasing trends in Shaanxi and Gansu provinces and a moderate increase in
Qinghai province. In Xinjiang and Ningxia where the most energy industries were
relocated and developed for the last decade (2005-2015), as aforementioned, $UF_k$
time series estimated using $SO_2$ VCD data illustrate clear upward trends. Compared
with those well developed regions in eastern and southern China, the $UF_k$ values of
$SO_2$ VCD in these northwestern provinces are almost all positive, except for Shaanxi
province where the $UF_k$ turned to negative from 2008,and Gansu province where
the $UF_k$ value become negative during 2012-2013.

The step change points identified by the MK test for $SO_2$ VCD in northwestern

China appear associated strongly with the development and use of coal energy. As
shown in **Fig. 9**, the intersection of the forward and backward sequences $UF_k$ and
$UB_k$ within the confidence levels of -1.96 (straight green line) to 1.96 (straight
purple line) can be identified in 2006 and 2007 in Ningxia and Xinjiang, respectively,
corresponding well to the expansion of two largest energy industry bases from 2003
onward in Ningxia (NECIB) and Midong energy industry park in Xinjiang. The step
change point of $SO_2$ VCD in 2012 in Gansu province coincides with fuel-switching
from coal to gas in the capital city (Lanzhou) and many other places of the province



initiated from 2012 (CSC, 2013b). The MK derived step change point in Shaanxi
province occurs in 2010 which is a clear signal of marked decline of fossil fuel
products in northern Shaanxi where, as the part of the EGT (Ma and Xu, 2017) of
China, the largest energy industry base in the province is located, right after the
global financial crisis.

It is interesting to note that the forward sequences $UF_k$ of $SO_2$ VCD (**Fig. 9e** and

**f**) in Ningxia and Xinjiang exhibit the similar fluctuations as that in Ningdong
(NECIB) and Midong energy industrial bases (**Fig. 8e** and **f**), manifesting the
potential associations between the $SO_2$ emissions in these two large-scale energy
industrial parks (major point sources) and provincial emissions in Ningxia and
Xinjiang, respectively. This suggests that large-scale energy industrial parks and bases
might likely overwhelm or play an important role in the $SO_2$ emissions in those
energy-abundant provinces in northwestern China. To assess the connections between
the major point sources in the two energy industrial parks and the provincial
emissions, we made use of OMI measured $SO_2$ VCD to inversely simulate the $SO_2$
emission burdens in Xinjiang and Ningxia. We used the source detection algorithm
(McLinden et al., 2016) and the approach, which fits OMI-measured $SO_2$ vertical
column densities to a three-dimensional parameterization function of the horizontal
coordinates and wind speed, proposed by Fioletov et al. (2015, 2016), to estimate the
$SO_2$ source strength in the two industrial parks and its contribution to the provincial
total $SO_2$ burdens. **Figure 10** illustrates mean $SO_2$ burdens from 2005 to 2015 in
northern Xinjiang (**Fig. 10a**) and Ningxia (**Fig. 10b**). The largest burdens can be seen



clearly in the Midong energy industrial base and the NECIB in these two minority
autonomous regions of China. Lower $SO_2$ emission burdens are illustrated in
mountainous areas of northern Xinjiang. **Figure 11** illustrates the annual variations of
estimated $SO_2$ emission burdens ($10^{26}$ molecules) in the NECIB and Midong energy
industrial parks (scaled on the left Y axis) and their respective fractions (%, scaled on
the right Y axis) in the total provincial $SO_2$ burdens in Ningxia and Xinjiang,
respectively. The $SO_2$ burden increased from 2005 and reached the maximum in 2011
in the NECIB and declined thereafter, in line with the annual $SO_2$ VCD fluctuations
(**Fig. 5**) in this industrial park which is, as aforementioned, attributable to the
economic rebound in 2011 in China. Of particular interest is the large fraction of the
estimated $SO_2$ emission burden in the NECIB in Ningxia (**Fig. 11a**), showing that this
industrial park alone contributed to about 40-50% emission burdens to the provincial
total $SO_2$ emission burden. Likewise, the $SO_2$ emission burden enhanced from 2005
and peaked in 2013 in Midong energy industrial park (**Fig. 11b**). The emission burden
in this park contributed about 25-35% to the provincial total $SO_2$ emission burden.
Compared with the NECIB, the $SO_2$ emission burden is higher in the Midong
industrial park but has the lower fraction in the provincial total emission burden.
Covered by large area of desert and Gobi (Junggar Basin) underlying surfaces, there
are only a few of $SO_2$ emission sources in vast northern Xinjiang region (total area of
Xinjiang is $1.66 \times 10^6$ km$^2$), leading to the small ratio of the major point source
(Midong) to total emission sources in Xinjiang. Nevertheless, overall our results
manifest that, although there were only a small number of $SO_2$ point sources in these





two energy industrial parks, the SO$_2$ emissions from these parks made significant
contributions to provincial total emissions. Given that the national strategy for China's
energy expansion and safety during the 21st century is, to a large extent, to develop
large scale energy industrial parks in northwestern China, particularly in Xinjiang and
Ningxia (Zhu and Ruth, 2015; Chen et al., 2016) where the energy resources are most
abundant in China, we would expect that the rising SO$_2$ emissions in northwestern
China would increasingly be attributed to those large scale energy industrial parks and
contributed increasingly to the national total SO$_2$ emission in China.
**Table 1** presents the annual average growth rates of SO$_2$ VCD, industrial
(second) Gross Domestic Product (GDP), and major coal-consuming industries in
northwestern China and three developed areas (BTH, YRD, PRD) in eastern and
southern China. The positive growth rates of SO$_2$ VCD can be observed in the three
province and autonomous regions (Qinghai, Ningxia, and Xinjiang) of northwestern
China. Although the growth rates of SO$_2$ VCD in other two provinces (Gansu and
Shaanxi) are negative, the magnitudes of the negative growth rates are smaller than
those in the BTH, YRD, and PRD, except for Zhejiang province in the YRD. This
regional contrast reflects both their economic and energy development activities, and
the SO$_2$ emission control measures implemented by the local and central
governments of China. Although China has set a national target of 10% SO$_2$
emission reduction (relative to 2005) during 2006-2010 and 8% (relative to 2010)
during 2011-2015 (CSC, 2007; CSC, 2011b), under the Grand Western Development
Program of China,the regulation for SO$_2$ emission control was waived in those





energy-abundant provinces of northwestern China in order to speed up the large
scale energy industrial bases and local economic development, and improve local
personal income. In addition, although FGDs were widely installed in coal-fired
power plants and other industrial sectors since the 1990s, by 2010 as much as 57%
of these systems were installed in eastern and southern China (Zhao et al., 2013).
The capacity of small power generators which were shut-down in western China was
merely about 10808 MW, only accounting for about 19% of the capacity of total
small power plants which were eliminated in China (55630 MW) during the 11th
Five-Year Plan period (2006-2010) (Cui et al., 2016). As shown in **Table 1**, the $SO_2$
emission reduction plans virtually specified the zero percentage of $SO_2$ emission
reductions in Qinghai, Gansu, and Xinjiang and lower reduction percentage in the
emission reduction in Ningxia and Inner Mongolia as compared to eastern and
southern China during the 11th (2006-2010) and 12th (2011-2015) Five-Year Plan.
As a result, the average growth rate for thermal power generation, steel production,
and coal consumption from 2005 to 2015 in northwestern China reached 14.1% $yr^{-1}$,
35.7% $yr^{-1}$, and 11.9% $yr^{-1}$, considerably higher than the averaged growth rates over
eastern and southern China (5.9% $yr^{-1}$ in the BTH,, 0.8% $yr^{-1}$ in the YRD, and 2.3%
$yr^{-1}$ in the PRD).

**4 Conclusions**

The spatiotemporal variation in $SO_2$ concentration during 2005-2015 over

China was investigated by making use of the PBL $SO_2$ column concentrations



measured by the Ozone Monitoring Instrument. The highest $SO_2$ VCD was found in
the NCP, the most heavily polluted area by $SO_2$ and particular matters (PM) in China,
including Beijing-Tianjin-Hebei, Shandong, and Henan province. Under the national
regulation for $SO_2$ control and emission reduction, the $SO_2$ VCD in eastern and
southern China underwent widespread decline during this period. However, the OMI
measured $SO_2$ VCD detected two "hot spots" in the EGT (Ningxia-Shaanxi-Inner
Mongolia) and Midong (Xinjiang) energy industrial parks, in contrast to the
declining $SO_2$ emissions in eastern and southern China, displaying an increasing
trend with the annual growth rate of 3.4% $yr^{-1}$ in the EGT and 1.8% $yr^{-1}$ in Midong,
respectively. The trend analysis further revealed enhanced $SO_2$ emissions in most
provinces of northwestern China likely due to national strategy for energy industry
expansion and relocation in energy-abundant northwestern China. As a result, per
capita $SO_2$ emission in northwestern China has exceeded industrialized and
populated eastern and southern China, making increasing contributions to the
national total $SO_2$ emission. The estimated $SO_2$ emission burdens in the Ningdong
(Ningxia) and Midong (Xinjiang) energy industrial parks from OMI measured $SO_2$
VCD showed that the $SO_2$ emissions in these two industrial parks made significant
contributions to the provincial total emissions. This indicates, on one side, that the
growing $SO_2$ emissions in northwestern China would increasingly come from those
large scale energy industrial parks under the national energy development and
relocation plan. On the other side, this fact also suggests that it is likely more
straightforward to control and reduce $SO_2$ emissions in northwestern China because



the SO$_2$ control measures could be readily implemented and authorized in those
state-owned large-scale energy industrial bases.

**The Supplement related to this article is available online**
*Acknowledgements.* This work is supported by the National Natural Science
Foundation of China (grants 41503089, 41371478, and 41671460), Gansu Province
Science and Technology Program for Livelihood of the People (1503FCMA003), the
Natural Science Foundation of Gansu Province of China (1506RJZA212), and
Fundamental Research Funds for the Central Universities (lzujbky-2016-249 and
lzujbky-2016-253). We thank Dr. Vitali Fioletov for his suggestions and advices
during the course of preparation of this manuscript.

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





**Table 1** Annual growth rate for OMI SO$_2$ VCD and economic activities for
individual provinces and municipality during 2005-2014 (%yr$^{-1}$), and SO$_2$ emission
reduction plan during the 11th and 12th Five-Year Plan period (%).

| Region | | OMI SO$_2$ VCD | coal consumption | Industrial GDP | Thermal power generation | steel production | SO$_2$ emission reduction plan (%) | |
|---|---|---|---|---|---|---|---|---|
| | | | | | | | 2006-2010[a] | 2011-2015[b] |
| Northwestern | Inner Mongolia | 0.94 | 11.29 | 20.48 | 14.07 | 8.38 | -3.8 | -3.8 |
| | Shaanxi | -3.41 | 13.14 | 19.96 | 13.01 | 14.48 | -12 | -7.9 |
| | Gansu | -0.09 | 6.69 | 14.19 | 8.89 | 9.92 | 0 | 2.0 |
| | Qinghai | 0.69 | 11.20 | 18.70 | 9.88 | 12.37 | 0 | 16.7 |
| | Ningxia | 0.95 | 11.79 | 17.44 | 15.04 | 152.71 | -9.3 | -3.6 |
| | Xinjiang | 1.57 | 17.21 | 14.21 | 23.39 | 16.27 | 0 | 0 |
| BTH | Beijing | -3.59 | -6.13 | 9.13 | 5.99 | -48.52 | -20.4 | -13.4 |
| | Tianjin | -4.63 | 3.15 | 15.84 | 6.01 | 10.19 | -9.4 | -9.4 |
| | Hebei | -5.05 | 4.16 | 12.37 | 6.22 | 10.70 | -15 | -12.7 |
| YRD | Shanghai | -7.65 | -0.93 | 6.64 | 0.86 | -0.92 | -26.9 | -13.7 |
| | Jiangsu | -5.93 | 5.39 | 12.51 | 7.49 | 13.35 | -18.0 | -14.8 |
| | Zhejiang | -2.07 | 4.04 | 11.40 | 8.68 | 13.94 | -15.0 | -13.3 |
| PRD | Guangdong | -4.55 | 6.15 | 12.03 | 5.92 | 6.87 | -15.0 | -14.8 |

a and b represents proposed reduction in SO$_2$ emission in 2010 relative to 2005, and 2015 relative
to 2010, respectively. The value for PRD refers to the proposed target for Guangdong Province.















**Figure Captions**

**Figure 1** Provinces, autonomous regions, and selected regions in China in this investigation. Northwestern China, defined by pink slash, includes Inner Mongolia, Shaanxi, Gansu, Qinghai, Ningxia, and Xinjiang province. Light green shadings with cross highlight Beijing-Tianjin-Hebei (BTH) and the light green color stands for the North China Plain (NCP, including BTH), defined by light green color, including BTH, Shandong, and Henan province. The Sichuan Basin, Yangtze River Delta (YRD), and Pearl River Delta (PRD) is defined by yellow, pink, and blue color. Red triangle indicate 188 monitoring sites across China.

**Figure 2** Annually averaged $SO_2$ VCD (DU), scaled on the right-hand-side Y-axis and measured annual $SO_2$ air concentration ($\mu g/m^3$), scaled on the left-hand-side Y-axis, in Beijing, Shanghai, Chongqing, Guangzhou, Yinchuan, and Urumqi.

**Figure 3** Annually averaged SO2 VCD (DU), scaled on the right-hand-side Y-axis and annual emissions (thousand ton/yr) of SO2 on the left-hand-side Y-axis in the NCP, YRD, PRD, Sichuan Basin, EGT, and Midong.

**Figure 4** Annual averaging OMI-retrieved vertical column densities of $SO_2$ (DU) and their trends from 2005 to 2015 on $0.25° \times 0.25°$ latitude/longitude resolution in China. (**a**). Annual mean $SO_2$ vertical column densities; (**b**). slope (trend) of linear regression relationship between annual average OMI-retrieved $SO_2$ VCD and the time sequence from 2005 to 2015 over China. The positive values indicate an increasing trend of $SO_2$ VCD from 2005 to 2015, and vice versa. The blue circle highlights the six selected regions where $SO_2$ VCD displayed dramatic change for further assessment of the long term trends and step change points in $SO_2$ VCD. These six regions are NCP (a), YRD (b), PRD (c), Sichuan Basin (d), Energy Golden Triangle (EGT, e), and Midong (f).

**Figure 5** Percentage changes in annual mean OMI $SO_2$ VCD in the four highlighted regions in eastern and southern China and two large-scale energy industry parks in the EGT and Midong region in **Figure 4b** (relative to 2005).

**Figure 6** Annual fractions of OMI retrieved $SO_2$ VCD and emissions averaged over 6 northwestern provinces in the national total $SO_2$ VCD from 2005 to 2015 and emission from 2005 to 2014. (**a**) fraction of annual mean $SO_2$ VCD; (**b**) fraction of annual mean emission. Fractions of $SO_2$ VCD are calculated as the ratio of the sum of annually averaged $SO_2$ VCD in northwestern China to the sum of annually averaged $SO_2$ VCD in the national total from 2005 to 2015 (%).

**Figure 7** Per capita $SO_2$ emission in six provinces of northwestern China and three key eastern regions (tons/person). The value for PRD refers to the per capita $SO_2$ emission for Guangdong province.

**Figure 8** Mann-Kendall (MK) test statistics for annually $SO_2$ VCD in those highlighted regions (**Figs. 1** and **4b)** from 2005-2015. The blue solid line is the forward sequence $UF_k$ and the red solid line is the backward sequence $UB_k$ defined by Eq (5). The positive values for $UF_k$ indicate an increasing trend of $SO_2$ VCD, and vice versa. Two straight solid lines stand for confidence interval between -1.96 (straight green line) and 1.96 (straight purple line) in the MK test. The bold black line in the middle highlights zero value of $UF_k$ and $UB_k$. The bold black line in the





middle highlights zero value of $UF_k$ and $UB_k$. The intersection of $UF_k$ and $UB_k$
sequences within the intervals between two confidence levels indicates a step change
point.
**Figure 9** Mann-Kendall (MK) test statistics for annually $SO_2$ VCD in six provinces
in northwestern China from 2005-2015. The blue solid line is the forward sequence
$UF_k$ and the red solid line is the backward sequence $UB_k$ defined by Eq (5). The
positive values for $UF_k$ indicate an increasing trend of $SO_2$ VCD, and vice versa.
Two straight solid lines stand for confidence interval between -1.96 (straight green
line) and 1.96 (straight purple line) in the MK test. The intersection of $UF_k$ and $UB_k$
sequences within intervals between two confidence levels indicates a step change
point.
**Figure 10** Mean $SO_2$ burden estimated by the OMI measured $SO_2$ VCD (DU) using a
new emission detection algorithm (Fioletov et al., 2016). (**a**) $SO_2$ burden in northern
Xinjiang; (**b**) $SO_2$ burden in Ningxia.
**Figure 11** Annual $SO_2$ burdens ($10^{26}$ molecule) in the Ningdong and Midong energy
industrial parks and their fractions in provincial total $SO_2$ burden. (**a**). $SO_2$ burden
(blue bar) in Ningdong and its fraction (red solid line) in the total provincial $SO_2$
burden in Ningxia; (**b**). $SO_2$ burden (blue bar) in Midong and its fraction (red solid
line) in the total provincial $SO_2$ burden in Xinjiang. The left Y-axis stands for $SO_2$
emission burden and the right Y-axis denotes the fraction (%).



Figure 1

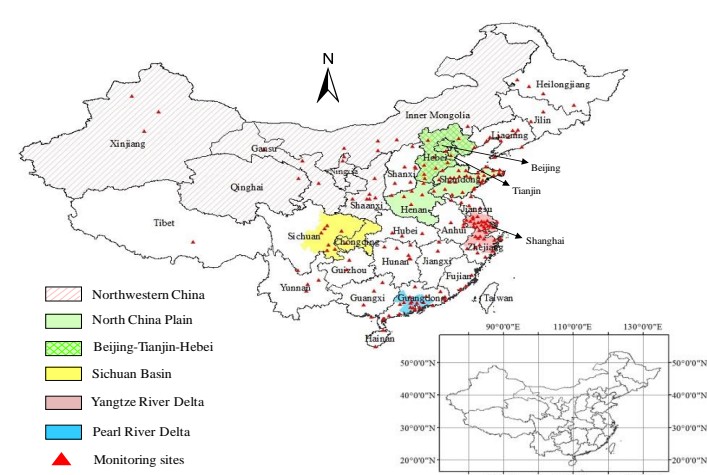






Figure 2

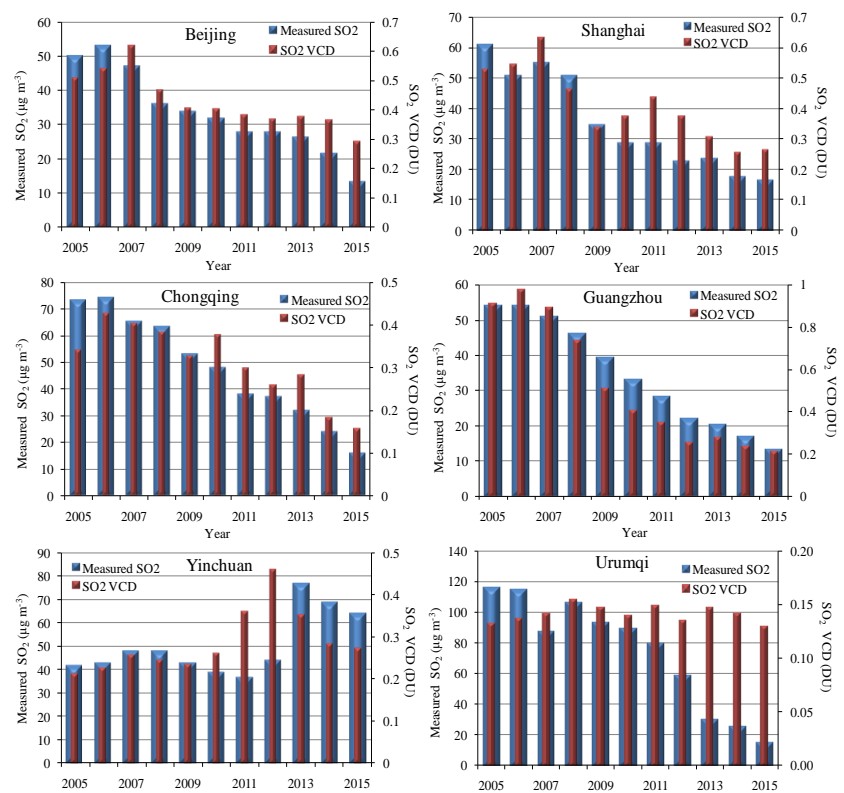























Figure 3

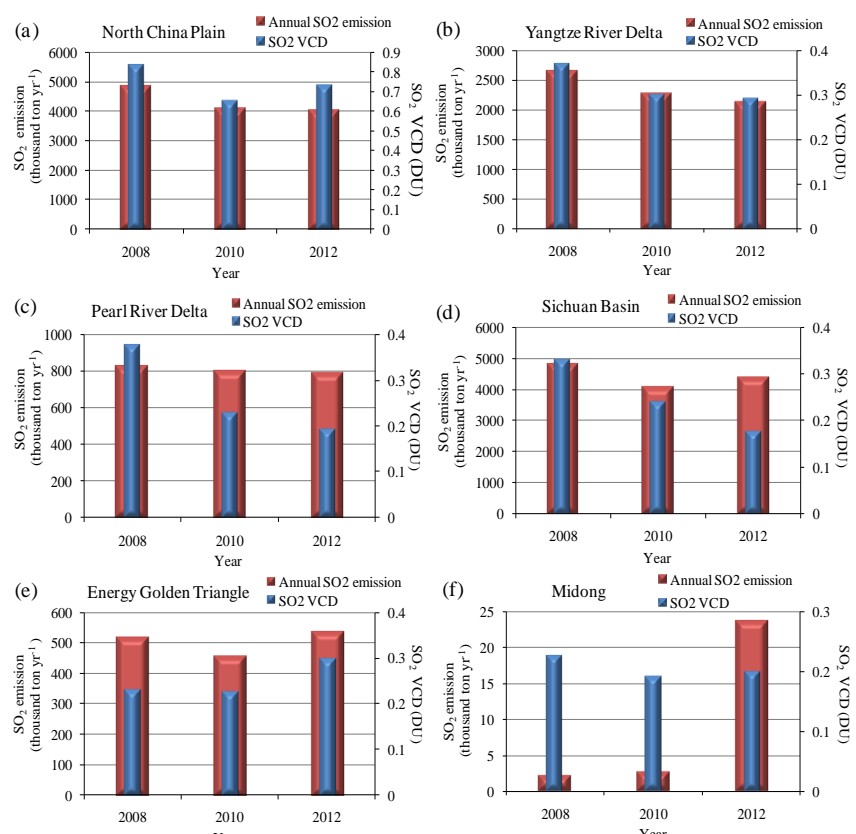




















Figure 4

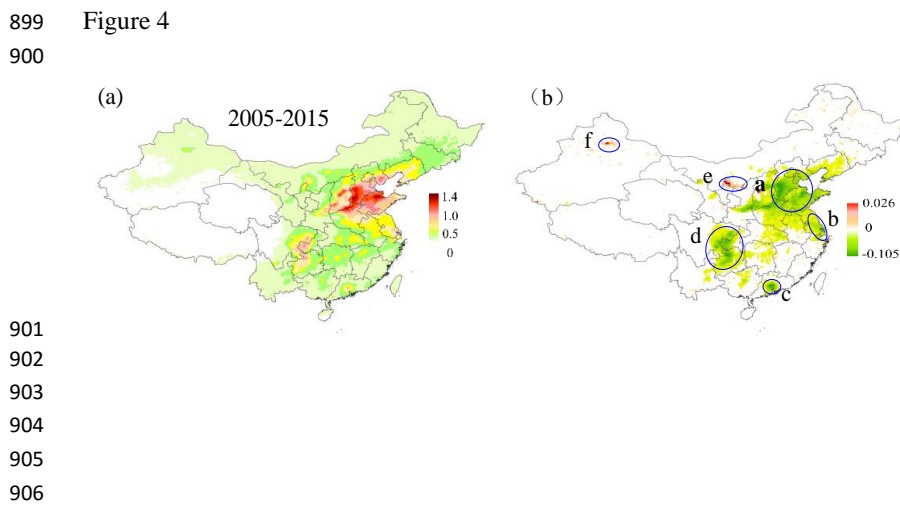

Figure 5

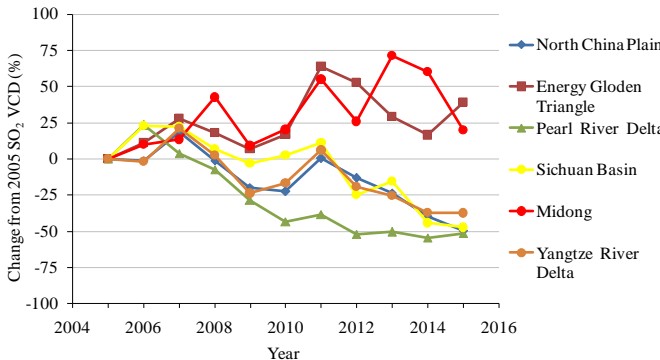






Figure 6

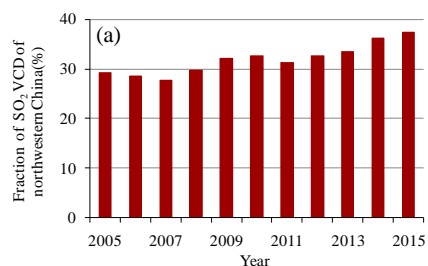 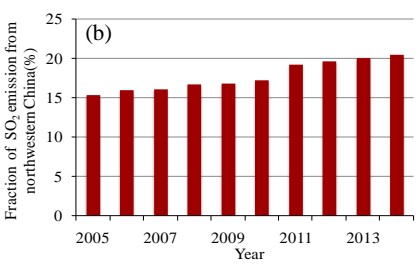











Figure 7

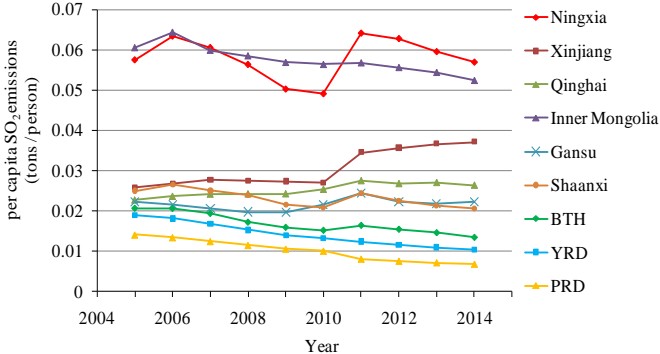














Figure 8

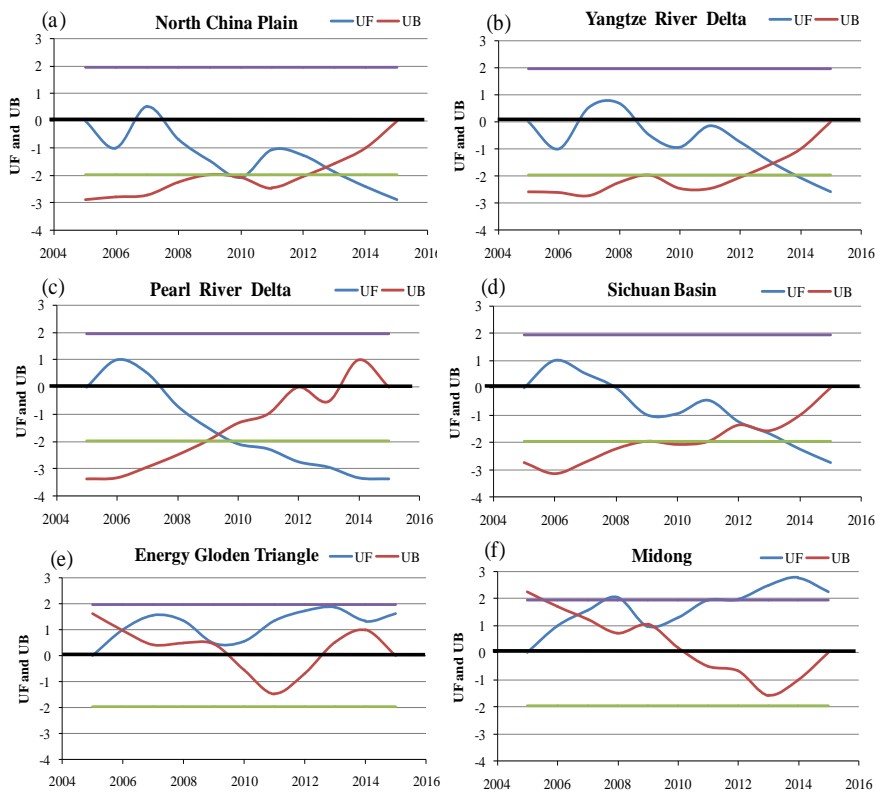






Figure 9

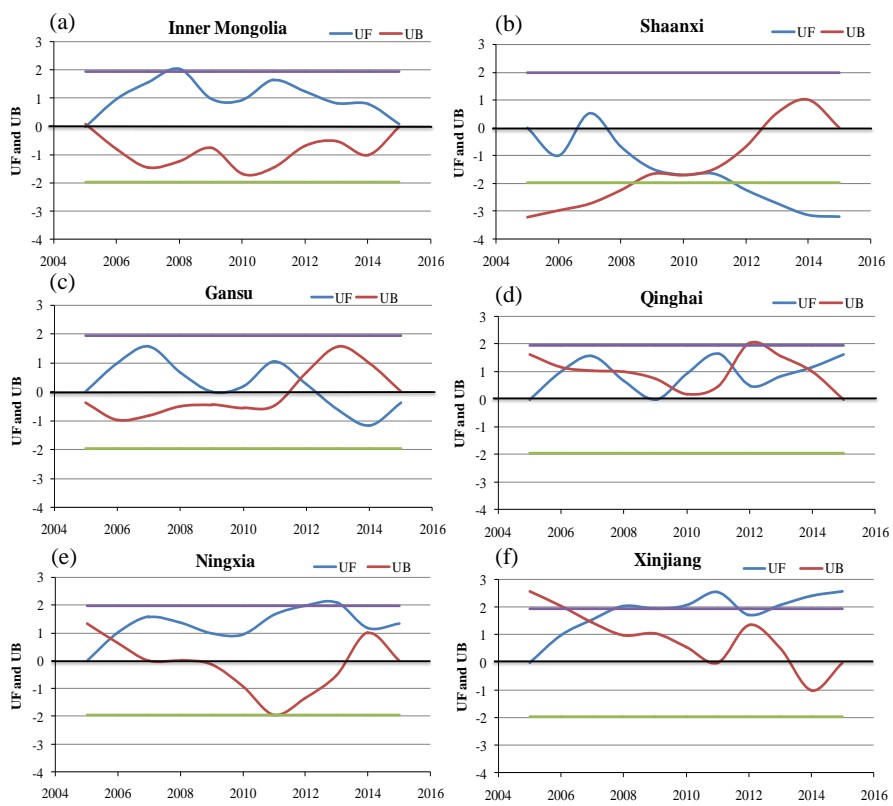























Figure 10

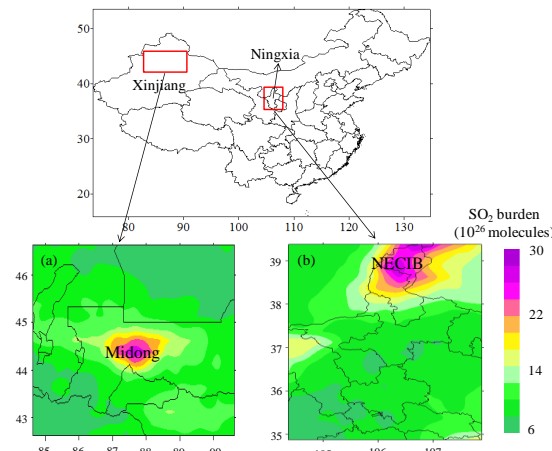









Figure 11

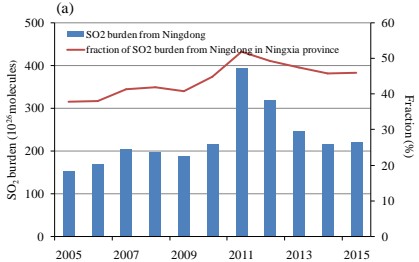
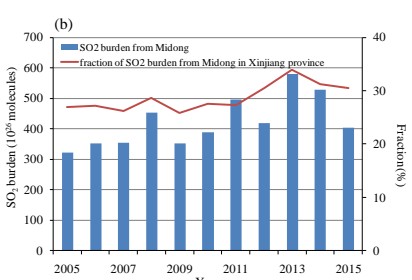
