# Peer review of "OMI measured increasing SO2 emissions due to energy industry"

_Atmospheric Chemistry and Physics, 2017_

## Referee Comment (RC1) · Anonymous Referee #2 · 6 May 2017

The manuscript discusses SO2 changes observed by OMI and links them to the national regulations of SO2 emissions. The paper demonstrates again the usefulness of satellite monitoring of air pollutions in China, the world largest SO2 emitter. It is shown that major changes in OMI records are linked to the emission reduction legislation. In general, the paper is well written, although some places require clarification. It can be published after minor revisions.

Comments

1. It is difficult to follow geographical names used by the authors. For example, Midong appears on p. 8, l. 145, without any mentioning of its location. As I understand, it is a district, but then the authors are talking about Urumqi-Midong region (p. 12, l. 241)

and Midong industrial park. Give more information about the cities and regions, provide cities coordinates, show all cities from Figure 2 in Figure 1.

2. P.7, l. 117, Figure 2. There is an explanation why the Urumqi plot is different from the others. Note that the measured SO2 concentration at Urumqi is the highest among all cities shown in Figure 2, while the OMI VCD values are the lowest. It suggests that the monitoring stations are located very close to the emission source (a power plant south of Urumqi?) and the emissions are not very large. The SO2 VCD values of about 0.1 DU are close to the noise level. The emission source is probably not large enough to produce elevated SO2 values in OMI data.

3. P.8, l. 145, Figure 2. SO2 emissions shown in Figure 2 for Midong are under 25 kt per year. OMI is not sensitive enough to see such emission sources, its sensitivity level is 30-40 kt per year (Fioletov et al., 2016). If there is a OMI hotspot in the area, that it is likely that the emissions from the source responsible for that hotspot are not in the emission inventory.

4. P. 19, l. 388-393 and Figure 10. This part is not clear. Papers McLinden et al., 2016, and Fioletov et al., 2016, used OMI Level 2 data merged with the wind profiles to estimate emissions from point sources. As I understand, the authors used Level 3 gridded data. What wind data were used and how the time was determined for grid cells? What is actually shown in Figure 10? The legend is in molecules, i.e., it can be interpreted as total SO2 mass. The caption says that it is in DU. Or, is it the emission rate? If the authors estimated emissions, they should elaborate more on the results. Do the estimated emissions agree with the reported ones? Are there any other sources within the areas shows in the two squares of Figure 10? If so, why are they not on the plot?

5. P.19, l. 393 and p. 20, 398, also Figure 11. The authors are talking about "SO2 burthen" and then "SO2 emission burdens" both in molecules. Are these two terms the same? It they are in molecules, they represent the total mass integrated over an

area and it is more convenient to show them in tones. If they represent emissions, they should be in units of mass per unit of time. Something is missing here.

6. P. 35, Table 1. What are the units in the OMI SO2 VCD column? Are the values in % per year for all columns except the last two where the values are in % per 5 years? Please clarify.

---

## Referee Comment (RC2) · Anonymous Referee #1 · 11 May 2017

This study demonstrates an increasing trend in SO2 over the northwestern region in China, in contrast to a well-established decreasing trend already reported for Eastern China. Shen et al., 2016 presented similar results before, however, here, the authors perform regression analysis/MK test, and 'a source detection approach to derive source strengths' using OMI-derived SO2 column density. They also report 30-50

A more rigorous and thorough analysis is required to confirm that the OMI-retrieved SO2 column densities can be used to derive/estimate the increasing trend in SO2 emissions/concentrations over these regions. Here, authors use Level-3 SO2 data at a particular spatial resolution with a constant AMF of 0.35. I would suggest a more detailed and in-depth study using the satellite SO2 column density dataset; in terms of

AMFs, spatial resolutions, various data filtering methods, sampling, averaging etc. and its impact on the results demonstrated here. This sort of a scientific analysis is required in order to come within the scope of ACP (rather than describing the trend analysis and spatiotemporal pattern of SO2 sources). McLinden et al., Fioletov et al., and Krotkov et al. papers are good references for this. Also, two years of in situ data over 188 sites offer a valuable piece of information (for example, L134:138: representativeness issues should have been addressed/described more carefully) to further test/evaluate satellite data (in addition to the supplementary figure and table). Also, describe in detail how the uncertainties in various datasets impact the results.

Need to correct for grammatical mistakes throughout the paper (examples; L2: economic growth; L9: reduction of; L127: but the both; L133: such the inconsistence; L200: an significant; 412: desert and Gopi? ...). Also, loose/empty sentences, and repetitions should be corrected while revising the paper. Change 'SO2' to 'SO2' for all the figures.

L81:82: try avoiding the point no.2, you can mention that, however, it's already an established point?

Section 2.1: describe more details of satellite SO2 data, error sources etc. This is the most important part of this paper.

L101:118: Better if you describe figures and tables in the results section. Describe just the 'materials and methods' in this section.

L133:134 skeptical of in situ? So, first, describe the dataset, and associated errors, and then describe your figures/results in that context.

Column density and emissions are correlated (supplementary figure and table). However, describe briefly why there are not linearly related; also, cite some relevant papers relating column density to emissions and surface concentrations (for example, using atmospheric models).

[Figure]

L134:139: how about using higher resolutions to address the issues of representative-ness? Also, these are loose/empty sentences.

L150:153: Those publications report some uncertainty estimates; report them here; and describe your figure in that context; more carefully.

L153:156: revise/avoid this sentence.

L157:162: briefly mention the socioeconomic data? GDP?

why per capita emissions used?

Results and discussion section is disorganized throughout. For the results section, first describe the decreases in SO2 over eastern China (as already reported in earlier publications), and focus more on the northwestern region (regions with increasing trend; this is the novel aspect of this paper?) in a separate sub-section.

Figure 4: colorbar should have the units

L385:393: describe 'source detection approach' (describe vertical column vs 'burden'; 'emission burden' a rate?) in the method section more clearly; and describe Figure 10/11 here in the results section itself. Better to overlay the column density data in figure 11. Also, a map of column density possible in figure 10 to see it in the context of these burden maps?

L462: mention about Particulate Matter (PM) in the introduction section itself.

---

## Author Comment (AC1) · 25 Jun 2017

First of all, we would like to thank the reviewers for his/her comments and suggestions which significantly improve the presentations and interpretations in our revised manuscript. Based on the reviewer's comments, we have made major revisions to the manuscript. The revised manuscript and supporting information are attached to Supplement. The reviewers' original comments and our responses are as follows:

This study demonstrates an increasing trend in SO2 over the northwestern region in China, in contrast to a well-established decreasing trend already reported for Eastern China. Shen et al., 2016 presented similar results before, however, here, the au-

[Figure]

thors perform regression analysis/MK test, and 'a source detection approach to derive source strengths' using OMI-derived SO2 column density. They also report ï¡d30-50% contribution of SO2 emissions over the two northwestern regions from two energy industrial parks. This work can be accepted for publication upon addressing the following suggestions.

1. A more rigorous and thorough analysis is required to confirm that the OMI-retrieved SO2 column densities can be used to derive/estimate the increasing trend in SO2 emissions/concentrations over these regions. Here, authors use Level-3 SO2 data at a particular spatial resolution with a constant AMF of 0.35. I would suggest a more detailed and in-depth study using the satellite SO2 column density dataset; in terms of AMFs, spatial resolutions, various data filtering methods, sampling, averaging etc. and its impact on the results demonstrated here. This sort of a scientific analysis is required in order to come within the scope of ACP (rather than describing the trend analysis and spatiotemporal pattern of SO2 sources). McLinden et al., Fioletov et al., and Krotkov et al. papers are good references for this. Also, two years of in situ data over 188 sites offer a valuable piece of information (for example, L134:138: representativeness issues should have been addressed/described more carefully) to further test/evaluate satellite data (in addition to the supplementary figure and table). Also, describe in detail how the uncertainties in various datasets impact the results.

Response: As we stated in our paper (line 119-122), we used Level-3 SO2 data at a particular spatial resolution with a constant AMF of 0.36 but the SO2 column density was adjusted by AMF values in China. Following the Reviewer's suggestions, we have rephrased text regarding the satellite data applied in the present study. In revised section 2.1, we introduced more detailed descriptions of the source, spatial resolutions, and potential errors of satellite data (line 86-122). In new sections 2.4 and 2.5, we added more details in the source detection algorithm developed by McLinden et al. and Fioletov et al. The sources of errors in determining the overall uncertainty of the SO2 emission estimation as well as their impact on the results were discussed

(line 219-230). We further added the comments on the causes of the inconsistency between SO2 VCD and monitored data (line 257-278 of the revised manuscript). We have quantified the uncertainties in the SO2 emissions derived from OMI measurements in the two major point sources in northwestern China by running the source detection model repeatedly for 10,000 times using Monte Carlo method. Results show the standard deviation of -35 to 122 kt/yr for SO2 emissions in NECIB and -29 to 95 kt/yr for SO2 emissions in MEIB from 2005 to 2015 which are presented in Fig. 11a and b, respectively (line 219-230 of the revised manuscript)

2. Need to correct for grammatical mistakes throughout the paper (examples; L2: economic growth; L9: reduction of; L127: but the both; L133: such the inconsistence; L200: an significant; 412: desert and Gopi? : : :). Also, loose/empty sentences, and repetitions should be corrected while revising the paper. Change 'SO2' to 'SO2' for all the figures.

Response: We have made every effort to improve language and taken more careful proofreading of the revised manuscript. Those spells and language errors have been corrected (e.g. 'destert and gobi' changed to Gobi desert) . We have changed 'SO2' to 'SO2' in all the figures.

3. L81:82: try avoiding the point no.2, you can mention that, however, it's already an established point?

Response: We thank the Reviewer for his/her suggestion. We have rewritten the second objective of this paper as "identify main causes contributing to the enhanced SO2 emission in northwestern China" (line 81-82 of the revised manuscript).

4. Section 2.1: describe more details of satellite SO2 data, error sources etc. This is the most important part of this paper.

Response: As our above response to the Reviewer's comment, following the Reviewer's suggestions, we have rewritten the description of satellite data (section 2.1).

We also added the source, spatial resolutions, error, and uncertainties of satellite data used in China in this study in two new sections 2.4 and 2,5.

5. L101:118: Better if you describe figures and tables in the results section. Describe just the 'materials and methods' in this section.

Response: Following the Reviewer's suggestion, we have rearranged the structure of Data and Methods section. We added the new section 2.5 (satellite data validation), and moved the discussions on the results presented in Table S2, Figure S1. Figures 2 and 3 were presently presented in Supplement but moved to Data and Methods section following the suggestion from a reviewer.

6. L133:134 skeptical of in situ? So, first, describe the dataset, and associated errors, and then describe your figures/results in that context.

Response: Following the Reviewer's comments and suggestions, in the revised manuscript, we have analyzed the causes leading to the inconsistence between SO2 VCD and monitored data (line 257-278 of the revised manuscript).

7. Column density and emissions are correlated (supplementary figure and table). However, describe briefly why there are not linearly related; also, cite some relevant papers relating column density to emissions and surface concentrations (for example, using atmospheric models).

Response: We thank the Reviewer for his/her suggestion. We have conducted new analysis on the inconsistence between SO2 emission and satellite observations data (line 283-298 of the revised manuscript).

8. L134:139: how about using higher resolutions to address the issues of representativeness? Also, these are loose/empty sentences.

Response: We agree that higher resolutions can reduce errors between SO2 VCD and monitored data. However, given the unavailability of data, only annual average monitored SO2 concentration in Urumqi city can be collected from the official data, which

is spatially averaged concentration over several monitoring sites across the city. This disagreement is unlikely resulted from the spatial resolution of satellite and measured SO2 data because good agreements between SO2 VCD and monitored concentrations can be seen in other cities. As aforementioned, we have discussed the causes resulting in the inconsistence between SO2 VCD and monitored data in the revised manuscript (line 257-278).

9. L150:153: Those publications report some uncertainty estimates; report them here; and describe your figure in that context; more carefully.

Response: Following the Reviewer's suggestions. We have added the uncertainties of SO2 emission in China, and described Figure 3 (line 293-303 of the revised manuscript).

10. L153:156: revise/avoid this sentence.

Response: This sentence has been rephrased in the revised paper (line 303-306).

11. L157:162: briefly mention the socioeconomic data? GDP? why per capita emissions used?

Response: We have added the detail socioeconomic data in the revised manuscript (line 142-144). In general, higher SO2 emissions are reported in those populated and industrialized regions. The use of per capita emission was to highlight the significance of SO2 emission in northwestern China and the fairness in accounting for SO2 emissions across China.

12. Results and discussion section is disorganized throughout. For the results section, first describe the decreases in SO2 over eastern China (as already reported in earlier publications), and focus more on the northwestern region (regions with increasing trend; this is the novel aspect of this paper?) in a separate sub-section.

Response: Following the Reviewer's suggestion, we have reorganized Results and Discussion section. In subsection 3.1 'OMI measured SO2 in China', we briefly dis-
cussed spatial-temporal distribution and fluctuations of SO2 VCD in China with focus on eastern and southern China. In subsection 3.2 'OMI measured SO2 'hot spots' in northwestern China', we highlighted two SO2 contaminated 'hot spots' featured by increasing SO2 VCDs in two large-scale energy industrial bases. In subsection 3.3 'OMI SO2 time series and step change point year in northwestern China', we extended our discussions and analysis from the increasing SO2 VCD in the two 'hot spots' to entire northwestern China which might be linked with SO2 emissions in those energy industrial bases. To be consistence with the new paper flow in the section, we moved Fig. 8 to subsection 3.1 as Fig. 6.

13. Figure 4: color bar should have the units.

Response: Done!

14. L385:393: describe 'source detection approach' (describe vertical column vs 'burden'; 'emission burden' a rate?) in the method section more clearly; and describe Figure 10/11 here in the results section itself. Better to overlay the column density data in figure 11. Also, a map of column density possible in figure 10 to see it in the context of these burden maps?

Response: Detailed source detection approach has been added to Date and Method section in the revised paper. We also presented detailed descriptions of SO2 emission estimate in new section 2.4. There was an error in previous Fig. 10. In figure caption and corresponding discussions we talked about SO2 emission burden. In the revised paper Fig. 10 shows SO2 VCD. Corresponding discussions were also revised (line 487-494). The estimated SO2 emissions using the source detection algorithm (Fioletove et al. 2015, 2016), VCDs, and their respective fractions are illustrated in revised Fig. 11.

15. L462: mention about Particulate Matter (PM) in the introduction section itself.

Response: We have deleted this phrase.

Please also note the supplement to this comment:
http://www.atmos-chem-phys-discuss.net/acp-2017-161/acp-2017-161-AC1-supplement.zip
* * *

---

## Author Comment (AC2) · 25 Jun 2017

First of all, we would like to thank the reviewer for his/her comments and suggestions which significantly improve the presentations and interpretations in our revised manuscript. Based on the reviewers' comments, we have made major revisions to the manuscript. The revised manuscript and supporting information are attached to Supplement. The reviewers' original comments and our responses are as follows:

The manuscript discusses SO2 changes observed by OMI and links them to the national regulations of SO2 emissions. The paper demonstrates again the usefulness of satellite monitoring of air pollutions in China, the world largest SO2 emitter. It is shown

that major changes in OMI records are linked to the emission reduction legislation. In general, the paper is well written, although some places require clarification. It can be published after minor revisions.

1. It is difficult to follow geographical names used by the authors. For example, Midong appears on p. 8, l. 145, without any mentioning of its location. As I understand, it is a district, but then the authors are talking about Urumqi-Midong region (p. 12, l. 241) and Midong industrial park. Give more information about the cities and regions, provide cities coordinates, show all cities from Figure 2 in Figure 1.

Response: We have revised Figure 1. We also added the selected cities shown in Fig. 2 to Figure 1, and marked several "hot spots" regions, including Urumqi-Midong region and Energy Golden Triangle (EGT), Ningdong energy chemical industrial base (NECIB), and Midong energy industrial base (MEIB), in northwestern China in Figure 1.

2. P.7, l. 117, Figure 2. There is an explanation why the Urumqi plot is different from the others. Note that the measured SO2 concentration at Urumqi is the highest among all cities shown in Figure 2, while the OMI VCD values are the lowest. It suggests that the monitoring stations are located very close to the emission source (a power plant south of Urumqi?) and the emissions are not very large. The SO2 VCD values of about 0.1 DU are close to the noise level. The emission source is probably not large enough to produce elevated SO2 values in OMI data.

Response: The measured SO2 concentration in Urumqi is the highest among all cities as shown in Fig. 2. However, as the Reviewer noted, the OMI SO2 VCD value in Urumqi was lower than other selected cities. This may be due to the error from systematic biases in OMI-retrieved SO2 VCD. Here we used the level 3 OMI PBL SO2 VCD data produced by the PCA retrievals to estimate the spatiotemporal variation in SO2 pollution in China. The PCA retrievals have a negative bias over some highly reflective surfaces such as many places in the Sahara (up to -0.5 DU in monthly mean).

The systematic bias of PCA retrieval is estimated at ∼0.5 DU for regions between 30°S and 30°N and ∼0.7-0.9 DU in relatively high latitude regions. Located in northwestern China and covered by Gobi desert in the surrounding regions of Urumqi, lower SO2 VCD might be yielded by the PCA retrieval over Urumqi compared with other cities (line 264-278). This point has been added to the revised paper.

3. P.8, l. 145, Figure 2. SO2 emissions shown in Figure 2 for Midong are under 25 kt per year. OMI is not sensitive enough to see such emission sources, its sensitivity level is 30-40 kt per year (Fioletov et al., 2016). If there is a OMI hotspot in the area, that it is likely that the emissions from the source responsible for that hotspot are not in the emission inventory.

Response: We agree with the Reviewer's comments. As shown in Figure 3, the OMI measured SO2 VCD in Urumqi-Midong from 2008 to 2012 was approximately 0.2 DU that was comparable with that in the EGT. However, SO2 emission in Urumqi-Midong was only 4% of that in the EGT in 2012. In particular, SO2 emission in Urumqi-Midong was 0.5% of that in the EGT from 2008 to 2010. This is probably because SO2 emission sources were not reported in emission inventory. Atmospheric removal and advection processes may also contribute to the inconsistence between monitored and satellite observations. These arguments have been added to the revised manuscript (line 287-303).

4. P. 19, l. 388-393 and Figure 10. This part is not clear. Papers McLinden et al., 2016, and Fioletov et al., 2016, used OMI Level 2 data merged with the wind profiles to estimate emissions from point sources. As I understand, the authors used Level 3 gridded data. What wind data were used and how the time was determined for grid cells? What is actually shown in Figure 10? The legend is in molecules, i.e., it can be interpreted as total SO2 mass. The caption says that it is in DU. Or, is it the emission rate? If the authors estimated emissions, they should elaborate more on the results. Do the estimated emissions agree with the reported ones? Are there any other sources within the areas shows in the two squares of Figure 10? If so, why are they not on the

plot?

Response: We thank the Reviewer to point out this confusion. There was an error in previous Fig. 10. In old figure 10 caption and corresponding discussions we talked about SO2 emission burden. In the revised paper Fig. 10 shows SO2 VCD. Corresponding discussions were also revised (line 487-494). The estimated SO2 emissions using the source detection algorithm (Fioletove et al. 2015, 2016), VCDs, and their respective fractions are illustrated in revised Fig. 11. In a new subsection 2.4, we presented the details of SO2 emission estimate using the source detection algorithm developed by Fioletov et al. (2015, 2016) in which wind speed data were used.

We estimated the SO2 burden (in number of molecules in 1026) which represents the total SO2 mass. Again we thank the reviewer to indicate the error in the unit of SO2 burden. Now the revised Fig. 10 shows SO2 VCD with the unit of DU. Revised Fig. 11 shows the estimated SO2 emission with the unit of kt/yr (Fig. 11a and b) and VCD with the unit of DU (Fig. 11c and d) in MEIB and NECIB, respectively.

5. P.19, l. 393 and p. 20, 398, also Figure 11. The authors are talking about "SO2 burthen" and then "SO2 emission burdens" both in molecules. Are these two terms the same? It they are in molecules, they represent the total mass integrated over an area and it is more convenient to show them in tones. If they represent emissions, they should be in units of mass per unit of time. Something is missing here.

Response: Please see our last response to the Reviewer. Revised Fig. 11 now illustrates the estimated SO2 emission (Fig. 11a and b) and VCD (Fig. 11c and d) in MEIB and NECIB using the source detection algorithm. In text, "SO2 emission burdens" have been changed to "SO2 emission".

6. P. 35, Table 1. What are the units in the OMI SO2 VCD column? Are the values in % per year for all columns except the last two where the values are in % per 5 years? Please clarify.

Response: Table 1 presents the annual growth rate for OMI SO2 VCD and economic activities for individual provinces and municipality during 2005-2014 (% yr-1). For OMI SO2 VCD column, they represented annual growth rate of spatially averaged SO2 VCD in the individual regions. In Table 1, the last two columns represented SO2 emission reduction plan during the 11th and 12th Five-Year Plan period, released by Chinese government every five years.

Please also note the supplement to this comment:
http://www.atmos-chem-phys-discuss.net/acp-2017-161/acp-2017-161-AC2-supplement.zip

---

## Author Response (AR1)

**Revisions and responses to reviewers' comments**

First of all, we would like to thank the reviewers for their comments and suggestions which significantly improve the presentations and interpretations in our revised manuscript. Based on the reviewers' comments, we have made major revisions to the manuscript. The reviewers' original comments are shown in italics and our responses are given in normal fonts.

**Response to Anonymous Referee #1**

*This study demonstrates an increasing trend in $SO_2$ over the northwestern region in China, in contrast to a well-established decreasing trend already reported for Eastern China. Shen et al., 2016 presented similar results before, however, here, the authors perform regression analysis/MK test, and 'a source detection approach to derive source strengths' using OMI-derived $SO_2$ column density. They also report ∼ 30-50% contribution of $SO_2$ emissions over the two northwestern regions from two energy industrial parks. This work can be accepted for publication upon addressing the following suggestions.*

*1. A more rigorous and thorough analysis is required to confirm that the OMI-retrieved SO2 column densities can be used to derive/estimate the increasing trend in $SO_2$ emissions/concentrations over these regions. Here, authors use Level-3 $SO_2$ data at a particular spatial resolution with a constant AMF of 0.35. I would suggest a more detailed and in-depth study using the satellite $SO_2$ column density dataset; in terms of AMFs, spatial resolutions, various data filtering methods, sampling, averaging etc. and its impact on the results demonstrated here. This sort of a scientific analysis is required in order to come within the scope of ACP (rather than describing the trend analysis and spatiotemporal pattern of $SO_2$ sources). McLinden et al., Fioletov et al., and Krotkov et al. papers are good references for this. Also, two years of in situ data over 188 sites offer a valuable piece of information (for example, L134:138: representativeness issues should have been addressed/described more carefully) to further test/evaluate satellite data (in addition to the supplementary figure and table). Also, describe in detail how the uncertainties in various datasets impact the results.*

Response:

As we stated in our paper (line 119-122), we used Level-3 $SO_2$ data at a particular spatial resolution with a constant AMF of 0.36 but the $SO_2$ column density was adjusted by AMF values in China. Following the Reviewer's suggestions, we have rephrased text regarding the satellite data applied in the present study. In revised section 2.1, we introduced more detailed descriptions of the source, spatial resolutions, and potential errors of satellite data (line 86-122). In new sections 2.4 and

2.5, we added more details in the source detection algorithm developed by McLinden et al. and Fioletov et al. The sources of errors in determining the overall uncertainty of the $SO_2$ emission estimation as well as their impact on the results were discussed (line 219-230). We further added the comments on the causes of the inconsistency between $SO_2$ VCD and monitored data (line 257-278 of the revised manuscript).

We have quantified the uncertainties in the $SO_2$ emissions derived from OMI measurements in the two major point sources in northwestern China by running the source detection model repeatedly for 10,000 times using Monte Carlo method. Results show the standard deviation of -35 to 122 kt/yr for $SO_2$ emissions in NECIB and -29 to 95 kt/yr for $SO_2$ emissions in MEIB from 2005 to 2015 which are presented in Fig. 11a and b, respectively (line 219-230 of the revised manuscript)

*2. Need to correct for grammatical mistakes throughout the paper (examples; L2: economic growth; L9: reduction of; L127: but the both; L133: such the inconsistence; L200: an significant; 412: desert and Gopi? : : :). Also, loose/empty sentences, and repetitions should be corrected while revising the paper. Change 'SO₂' to 'SO₂' for all the figures.*

Response:

We have made every effort to improve language and taken more careful proofreading of the revised manuscript. Those spells and language errors have been corrected (e.g. 'destert and gobi' changed to Gobi desert) . We have changed 'SO2' to '$SO_2$' in all the figures.

*3. L81:82: try avoiding the point no.2, you can mention that, however, it's already an established point?*

Response:

We thank the Reviewer for his/her suggestion. We have rewritten the second objective of this paper as "identify main causes contributing to the enhanced $SO_2$ emission in northwestern China" (line 81-82 of the revised manuscript).

*4. Section 2.1: describe more details of satellite $SO_2$ data, error sources etc. This is the most important part of this paper.*

Response:

As our above response to the Reviewer's comment, following the Reviewer's suggestions, we have rewritten the description of satellite data (section 2.1). We also added the source, spatial resolutions, error, and uncertainties of satellite data used in China in this study in two new sections 2.4 and 2,5.

*5. L101:118: Better if you describe figures and tables in the results section. Describe just the 'materials and methods' in this section.*

Response:

Following the Reviewer's suggestion, we have rearranged the structure of Data and Methods section. We added the new section 2.5 (satellite data validation), and moved the discussions on the results presented in Table S2, Figure S1. Figures 2 and 3 were presently presented in Supplement but moved to Data and Methods section following the suggestion from a reviewer.

*6. L133:134 skeptical of in situ? So, first, describe the dataset, and associated errors, and then describe your figures/results in that context.*

Response:

Following the Reviewer's comments and suggestions, in the revised manuscript, we have analyzed the causes leading to the inconsistence between $SO_2$ VCD and monitored data (line 257-278 of the revised manuscript).

*7. Column density and emissions are correlated (supplementary figure and table). However, describe briefly why there are not linearly related; also, cite some relevant papers relating column density to emissions and surface concentrations (for example, using atmospheric models).*

Response:

We thank the Reviewer for his/her suggestion. We have conducted new analysis on the inconsistence between $SO_2$ emission and satellite observations data (line 283-298 of the revised manuscript).

*8. L134:139: how about using higher resolutions to address the issues of representativeness? Also, these are loose/empty sentences.*

Response:

We agree that higher resolutions can reduce errors between $SO_2$ VCD and monitored data. However, given the unavailability of data, only annual average monitored $SO_2$ concentration in Urumqi city can be collected from the official data, which is spatially averaged concentration over several monitoring sites across the city. This disagreement is unlikely resulted from the spatial resolution of satellite and measured $SO_2$ data because good agreements between $SO_2$ VCD and monitored concentrations can be seen in other cities. As aforementioned, we have discussed the causes resulting in the inconsistence between $SO_2$ VCD and monitored data in the revised manuscript (line 257-278).

*9. L150:153: Those publications report some uncertainty estimates; report them here; and describe your figure in that context; more carefully.*

Response:

Following the Reviewer's suggestions. We have added the uncertainties of $SO_2$ emission in China, and described Figure 3 (line 293-303 of the revised manuscript).

*10. L153:156: revise/avoid this sentence.*

Response:

This sentence has been rephrased in the revised paper (line 303-306).

*11. L157:162: briefly mention the socioeconomic data? GDP? why per capita emissions used?*

Response:

We have added the detail socioeconomic data in the revised manuscript (line 142-144). In general, higher $SO_2$ emissions are reported in those populated and industrialized regions. The use of per capita emission was to highlight the significance of $SO_2$ emission in northwestern China and the fairness in accounting for $SO_2$ emissions across China.

*12. Results and discussion section is disorganized throughout. For the results section, first describe the decreases in $SO_2$ over eastern China (as already reported in earlier publications), and focus more on the northwestern region (regions with increasing trend; this is the novel aspect of this paper?) in a separate sub-section.*

Response:

Following the Reviewer's suggestion, we have reorganized Results and Discussion section. In subsection 3.1 'OMI measured $SO_2$ in China', we briefly discussed spatial-temporal distribution and fluctuations of $SO_2$ VCD in China with focus on eastern and southern China. In subsection 3.2 'OMI measured $SO_2$ 'hot spots' in northwestern China', we highlighted two $SO_2$ contaminated 'hot spots' featured by increasing $SO_2$ VCDs in two large-scale energy industrial bases. In subsection 3.3 'OMI $SO_2$ time series and step change point year in northwestern China', we extended our discussions and analysis from the increasing $SO_2$ VCD in the two 'hot spots' to entire northwestern China which might be linked with $SO_2$ emissions in those energy industrial bases. To be consistence with the new paper flow in the section, we moved Fig. 8 to subsection 3.1 as Fig. 6.

*13. Figure 4: color bar should have the units.*

Response:

Done!

*14. L385:393: describe 'source detection approach' (describe vertical column vs 'burden'; 'emission burden' a rate?) in the method section more clearly; and describe Figure 10/11 here in the results section itself. Better to overlay the column density data in figure 11. Also, a map of column density possible in figure 10 to see it in the context of these burden maps?*

Response:

Detailed source detection approach has been added to Date and Method section in the revised paper. We also presented detailed descriptions of $SO_2$ emission estimate in new section 2.4. There was an error in previous Fig. 10. In figure caption and corresponding discussions we talked about $SO_2$ emission burden. In the revised paper Fig. 10 shows $SO_2$ VCD. Corresponding discussions were also revised (line 487-494). The estimated $SO_2$ emissions using the source detection algorithm (Fioletove et al. 2015, 2016), VCDs, and their respective fractions are illustrated in revised Fig. 11.

*15. L462: mention about Particulate Matter (PM) in the introduction section itself.*

Response:

We have deleted this phrase.

**Response to Anonymous Referee #2**

*The manuscript discusses $SO_2$ changes observed by OMI and links them to the national regulations of $SO_2$ emissions. The paper demonstrates again the usefulness of satellite monitoring of air pollutions in China, the world largest $SO_2$ emitter. It is shown that major changes in OMI records are linked to the emission reduction legislation. In general, the paper is well written, although some places require clarification. It can be published after minor revisions.*

*1. It is difficult to follow geographical names used by the authors. For example, Midong appears on p. 8, l. 145, without any mentioning of its location. As I understand, it is a district, but then the authors are talking about Urumqi-Midong region (p. 12, l. 241) and Midong industrial park. Give more information about the cities and regions, provide cities coordinates, show all cities from Figure 2 in Figure 1.*

Response:

We have revised Figure 1. We also added the selected cities shown in Fig. 2 to Figure 1, and marked several "hot spots" regions, including Urumqi-Midong region and Energy Golden Triangle (EGT), Ningdong energy chemical industrial base (NECIB), and Midong energy industrial base (MEIB), in northwestern China in Figure 1.

*2. P.7, l. 117, Figure 2. There is an explanation why the Urumqi plot is different from the others. Note that the measured $SO_2$ concentration at Urumqi is the highest among all cities shown in Figure 2, while the OMI VCD values are the lowest. It suggests that the monitoring stations are located very close to the emission source (a power plant south of Urumqi?) and the emissions are not very large. The $SO_2$ VCD values of about 0.1 DU are close to the noise level. The emission source is probably not large enough to produce elevated $SO_2$ values in OMI data.*

Response:

The measured $SO_2$ concentration in Urumqi is the highest among all cities as shown in Fig. 2. However, as the Reviewer noted, the OMI $SO_2$ VCD value in Urumqi was lower than other selected cities. This may be due to the error from systematic biases in OMI-retrieved $SO_2$ VCD. Here we used the level 3 OMI PBL $SO_2$ VCD data produced by the PCA retrievals to estimate the spatiotemporal variation in $SO_2$ pollution in China. The PCA retrievals have a negative bias over some highly reflective surfaces such as many places in the Sahara (up to -0.5 DU in monthly mean). The systematic bias of PCA retrieval is estimated at ~0.5 DU for regions between 30°S and 30°N and  ~0.7-0.9 DU in relatively high latitude regions. Located in northwestern China and covered by Gobi desert in the surrounding regions of Urumqi, lower $SO_2$ VCD might be yielded by the PCA retrieval over Urumqi compared with other cities (line 264-278). This point has been added to the revised paper.

*3. P.8, l. 145, Figure 2. SO2 emissions shown in Figure 2 for Midong are under 25 kt per year. OMI is not sensitive enough to see such emission sources, its sensitivity level is 30-40 kt per year (Fioletov et al., 2016). If there is a OMI hotspot in the area, that it is likely that the emissions from the source responsible for that hotspot are not in the emission inventory.*

Response:

We agree with the Reviewer's comments. As shown in Figure 3, the OMI measured $SO_2$ VCD in Urumqi-Midong from 2008 to 2012 was approximately 0.2 DU that was comparable with that in the EGT. However, $SO_2$ emission in Urumqi-Midong was only 4% of that in the EGT in 2012. In particular, $SO_2$ emission in Urumqi-Midong was 0.5% of that in the EGT from 2008 to 2010. This is probably because $SO_2$ emission sources were not reported in emission inventory. Atmospheric removal and advection processes may also contribute to the inconsistence between monitored and satellite observations. These arguments have been added to the revised manuscript (line 287-303).

*4. P. 19, l. 388-393 and Figure 10. This part is not clear. Papers McLinden et al., 2016, and Fioletov et al., 2016, used OMI Level 2 data merged with the wind profiles to estimate emissions from point sources. As I understand, the authors used Level 3 gridded data. What wind data were used and how the time was determined for grid cells? What is actually shown in Figure 10? The legend is in molecules, i.e., it can be interpreted as total SO2 mass. The caption says that it is in DU. Or, is it the emission rate? If the authors estimated emissions, they should elaborate more on the results. Do the estimated emissions agree with the reported ones? Are there any other sources within the areas shows in the two squares of Figure 10? If so, why are they not on the plot?*

Response:

We thank the Reviewer to point out this confusion. There was an error in previous Fig. 10. In old figure 10 caption and corresponding discussions we talked about $SO_2$ emission burden. In the revised paper Fig. 10 shows $SO_2$ VCD. Corresponding discussions were also revised (line 487-494). The estimated $SO_2$ emissions using the source detection algorithm (Fioletove et al. 2015, 2016), VCDs, and their respective fractions are illustrated in revised Fig. 11. In a new subsection 2.4, we presented the details of $SO_2$ emission estimate using the source detection algorithm developed by Fioletov et al. (2015, 2016) in which wind speed data were used.

We estimated the $SO_2$ burden (in number of molecules in $10^{26}$) which represents the total $SO_2$ mass. Again we thank the reviewer to indicate the error in the unit of $SO_2$ burden. Now the revised Fig. 10 shows $SO_2$ VCD with the unit of DU. Revised Fig. 11 shows the estimated $SO_2$ emission with the unit of kt/yr (Fig. 11a and b) and VCD with the unit of DU (Fig. 11c and d) in MEIB and NECIB, respectively.

*5. P.19, l. 393 and p. 20, 398, also Figure 11. The authors are talking about "$SO_2$ burthen" and then "$SO_2$ emission burdens" both in molecules. Are these two terms the same? It they are in molecules, they represent the total mass integrated over an area and it is more convenient to show them in tones. If they represent emissions, they should be in units of mass per unit of time. Something is missing here.*

Response:

Please see our last response to the Reviewer. Revised Fig. 11 now illustrates the estimated $SO_2$ emission (Fig. 11a and b) and VCD (Fig. 11c and d) in MEIB and NECIB using the source detection algorithm. In text, "$SO_2$ emission burdens" have been changed to "$SO_2$ emission".

*6. P. 35, Table 1. What are the units in the OMI $SO_2$ VCD column? Are the values in % per year for all columns except the last two where the values are in % per 5 years? Please clarify.*

Response:

Table 1 presents the annual growth rate for OMI $SO_2$ VCD and economic activities for individual provinces and municipality during 2005-2014 (% $yr^{-1}$). For OMI $SO_2$ VCD column, they represented annual growth rate of spatially averaged $SO_2$ VCD in the individual regions. In Table 1, the last two columns represented $
[revised manuscript text omitted]

[Figure]

Figure 2

[Figure]

Figure 3

[Figure]

Figure 42

[Figure]

[Figure]

Figure 53

[Figure]

[Figure]

Figure 4

[Figure]

Figure 5

[Figure]

Figure 6

[Figure]

-

Figure 7

[Figure]

[Figure]

Figure 8

[Figure]

[Figure]

[Figure]

Figure 9

[Figure]

[Figure]

Figure 10

[Figure]

Figure 11

[Figure]

---

## Author Response (AR2)

**Revisions and response to Co-Editor**

We would like to thank the Co-Editor for his suggestion which simplify figure captions for final publication. Based on the Co-Editor's suggestion, we have made revisions to the figure captions of manuscript. The marked-up manuscript was attached as next page.

[revised manuscript text omitted]

Figure 2

[Figure]

Figure 3

[Figure]

[Figure]

Figure 4

(a)    2005-2015    (b)

Figure 5

[Figure]

Figure 6

[Figure]

[Figure]

Figure 7

[Figure]

Figure 8

[Figure]

Figure 9

[Figure]

[Figure]

Figure 10

[Figure]

Figure 11

[Figure]